behaviour, ecology

*Apis mellifera*, propolis, *Varroa destructor*, natural pesticide, social medication

**Authors for correspondence:**
Alberto Satta
e-mail: albsatta@uniss.it
Francesco Nazzi
e-mail: francesco.nazzi@uniud.it

# Honeybees use propolis as a natural pesticide against their major ectoparasite

Michelina Pusceddu[1], Desiderato Annoscia[2], Ignazio Floris[1], Davide Frizzera[2], Virginia Zanni[2], Alberto Angioni[3], Alberto Satta[1] and Francesco Nazzi[2]

[1]Dipartimento di Agraria, Sezione di Patologia vegetale ed Entomologia, Università degli Studi di Sassari, Sassari, Italy
[2]Dipartimento di Scienze AgroAlimentari, Ambientali e Animali, Università degli Studi di Udine, Udine, Italy
[3]Dipartimento di Scienze della Vita e dell'Ambiente, Università degli Studi di Cagliari, Cagliari, Italy

MP, 0000-0003-0915-6258; DA, 0000-0002-2405-4914; IF, 0000-0003-1274-7147;
DF, 0000-0002-8809-2768; VZ, 0000-0001-6242-4608; AA, 0000-0001-7370-0903;
AS, 0000-0003-1296-3835; FN, 0000-0002-2085-7621

Honeybees use propolis collected from plants for coating the inner walls of their nest. This substance is also used as a natural antibiotic against microbial pathogens, similarly to many other animals exploiting natural products for self-medication. We carried out chemical analyses and laboratory bioassays to test if honeybees use propolis for social medication against their major ectoparasite: *Varroa destructor*. We found that propolis is applied to brood cells where it can affect the reproducing parasites, with a positive effect on honeybees and a potential impact on *Varroa* population. We conclude that propolis can be regarded as a natural pesticide used by the honeybee to limit a dangerous parasite. These findings significantly enlarge our understanding of behavioural immunity in animals and may have important implications for the management of the most important threat to honeybees worldwide.

## 1. Introduction

Propolis is a substance deriving from resins collected by foragers on plants, and used by honeybees to coat the walls of the cavity hosting their nest [1]. Propolis is also used to seal unwanted open spaces within the hive, to wrap the dead body of possible intruders and for polishing the honeycomb cells between successive brood cycles [2], although information about this latter use is scarce. Due to its antimicrobial properties, propolis can have a direct effect on several hive pathogens, including *Paenibacillus larvae* [3], *Ascosphaera apis* [4] and *Nosema ceranae* [5], that are the causal agents of American foulbrood, chalkbrood and nosemiasis, respectively. Furthermore, by reducing the hive's microbial load, propolis allow bees to invest less in individual immune function and may prime the detoxification pathways, with consequent beneficial effects at colony level [6]. For these reasons propolis is regarded as an important component of social medication in honeybees [7].

Since the recent shifting from the original host *Apis cerana* to *Apis mellifera*, the parasitic mite *Varroa destructor* has become the most serious threat to honeybees worldwide and plays a fundamental role in the decline of honeybee colonies observed in the Northern Hemisphere in the last decade [8]. The parasite causes a number of detrimental effects on bees at the individual level [9] and is involved in the transmission of bee viruses [10]. In particular, *Varroa* plays a central role in the transmission and activation of deformed wing virus (DWV), resulting in devastating overt infections [11]. The mite can parasitize both adult bees and pupae, but it is during the pupal stage, when reproduction takes place, that it causes most damage [8], such that eclosing bees often bear very high viral loads associated with crippled wings and drastically reduced lifespan [9]. Widespread resistance to commonly used acaricides poses a serious threat to efficient chemical control of the parasite [12] and makes

**Table 1.** Extraction rates (%) from two different types of honeycomb as obtained using hexane and MeOH/H$_2$O (80/20, v/v) solvents, and total phenols identified in the extracts.

| matrix | hexane % ± RSD | MeOH/H$_2$O % ± RSD | total phenols mg g$^{-1}$ ± RSD GAE eq |
|---|---|---|---|
| honeycomb under construction | 98.00 ± 5.24 | 1.48 ± 25.79 | 6.83 ± 14.87 |
| honeycomb prepared for the first oviposition | 92.80 ± 0.77 | 5.65 ± 10.01 | 152.22 ± 6.45 |
| propolis | — | 56.29 ± 7.17 | 119.06 ± 10.12 |

the development of novel methods urgent, particularly those targeting mites in the brood, thus not contributing to increase the parasite's virulence [13]. It has been noted that the proportion of resin foragers in bee colonies increases according to *Varroa* infestation [14], suggesting a causal link between parasitic challenge and propolis collection, consistent with increased survival of *Varroa*-infested worker bees reared in presence of propolis [15] or fed sugar syrup added with propolis [16]. It has also been observed that propolis reduces DWV viral loads associated with *V. destructor* infestations at colony level [17]. However, despite this interesting body of evidence regarding the possible role of propolis in social medication in *Varroa*-infested colonies, the possible underlying mechanisms still remain elusive. To fill this gap, we carried out experiments to assess if and how propolis is used by honeybees from mite-infested colonies. We tested the possible role of propolis both during pupation, when the mite feeds abundantly on bees and reproduces, and after eclosion, when the effect of parasitization results in a dramatic shortening of the bee's lifespan.

## 2. Results and discussion

To clarify if propolis is applied by bees to honeycomb cells and can thus act upon *Varroa* mites invading those cells for reproduction, we extracted honeycombs under construction or prepared for the first oviposition with different solvents. Honeycombs under construction were entirely dissolved by hexane, while they were not affected by a polar solvent (table 1). Instead, honeycombs prepared for the first oviposition were only partially dissolved in hexane, while the extraction rate of the alcoholic solution increased accordingly (table 1). The spectrophotometric analysis of total phenols from the alcoholic extracts obtained from the honeycombs ready for oviposition and propolis were fully comparable (table 1).

Based on the quantity of propolis extracted from a given weight of honeycomb, we estimated that the amount of resin spread on the inner walls of each cell is in the order of milligrams. Chemical analyses of combs extracts, propolis and diet solution, identified nearly 30 compounds, including flavones, flavonols, simple phenols, aglycones and conjugates aglycones (electronic supplementary material, table S1). The most abundant compounds in propolis were kaempferol and its homologues, accounting for almost 2 g kg$^{-1}$ of extract, whereas total phenols amounted to almost 120 g kg$^{-1}$ (electronic supplementary material, table S1). The comb extract showed a similar composition to propolis; however, some minor compounds were undetected, and others, such as cinnamic acid homologues, were more concentrated.

Overall, these results showed that, after construction, bees apply a phenol-rich propolis to the honeycomb.

To verify if propolis applied by bees to brood cells can influence *V. destructor* during the reproductive phase and to assess the resulting effect on the survival of its host *A. mellifera*, we reared honeybee larvae inside artificial cells, treated or not with an ethanolic extract of propolis (P+E+ and P−E−, respectively), in presence or not of a *Varroa* mite (V+ and V−, respectively). To this purpose we used gelatin capsules where the mite can reproduce at rates similar to natural cells [18]; the chemistry of this substrate is certainly different from the wax making the brood cells but does not differ much from the cocoon layer wrapping the inner surface of the brood cells where a bee larva has already developed.

Propolis significantly increased the mortality of *Varroa* mites, which approached 20% in treated cells (figure 1*a*; V+P+E+ versus V+P−E−: Mantel–Haenszel $\chi^2$: 10.156, d.f. = 1, $p$ = 0.001; V+P+E+ versus V+P−E+: Mantel–Haenszel $\chi^2$: 9.744, d.f. = 1, $p$ = 0.002), while ethanol alone had no effect (figure 1*a*; V+P−E+ versus V+P−E−; Mantel–Haenszel $\chi^2$: $1 \times 10^{-4}$, d.f. = 1, $p$ = 0.993). Also, propolis significantly decreased, by 44%, the percentage of surviving mites that produced offspring inside the rearing cells (figure 1*b*; V+P+E+ versus V+P−E+: Mantel–Haenszel $\chi^2$: 10.092, d.f. = 1, $p$ = 0.002; V+P+E+ versus V+P−E−: Mantel–Haenszel $\chi^2$: 13.280, d.f. = 1, $p$ < 0.001); again, ethanol alone had no effect (figure 1*b*; V+P−E+ versus V+P−E−, Mantel–Haenszel $\chi^2$: 0.099, d.f. = 1, $p$ = 0.753).

Mite infestation caused a significant increase in the mortality of bees developing inside the rearing cells, while propolis did not (electronic supplementary material, figure S1). Furthermore, mite infestation had a significant impact on the weight of surviving bees, but not propolis (electronic supplementary material, figure S2). These results indicate that propolis applied to brood cells before oviposition can influence the mites parasitizing the bee pupae, decreasing their survival and reducing reproduction, probably because of the acaricidal effects of propolis [15,16,19]. Such effects could be related to the interference with neuronal transmission [20] and/or reduced heat production and oxygen consumption [21]. Instead, at the pupal stage, propolis does not seem to be able to counteract the negative, direct effects of mite infestation on developing bees, as highlighted by the similar weight loss in bees from treated and untreated cells. At the emergence, a significantly higher proportion of bees from mite-infested untreated cells had crippled wing, a symptom consistently associated with very high viral infection levels (figure 2*a*; electronic supplementary material, table S2); interestingly, mite-infested bees from rearing cells treated with propolis were not significantly different from uninfested bees (figure 2*a*; electronic supplementary material, table S2). As expected, mite infestation significantly reduced

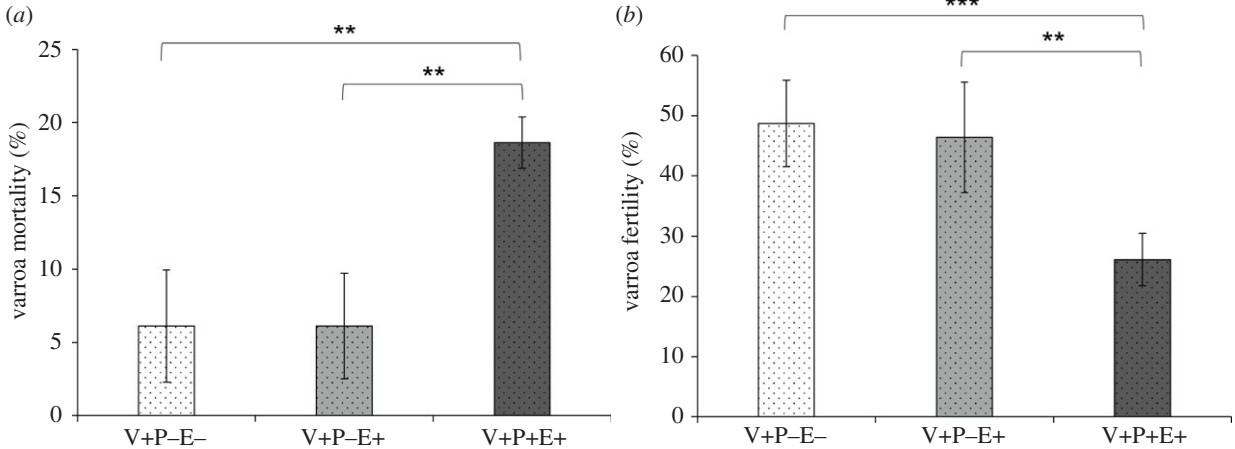

**Figure 1.** (*a*) Percentage mortality (mean ± s.d.) of *Varroa* mites reared in artificial cells treated with propolis ethanol extract (V+P+E+) or with ethanol only (V+P−E+, positive control) or left untreated (V+P−E−, negative control). (*b*) Percentage of surviving mites that produced offspring (i.e. fertility; mean ± s.d.) in artificial cells treated with an ethanolic extract of propolis (V+P+E+) or with ethanol only (V+P−E+, positive control) or left untreated (V+P−E−, negative control). Two asterisks mark experimental groups differing from each other at $p < 0.01$; three asterisks were used if $p < 0.001$.

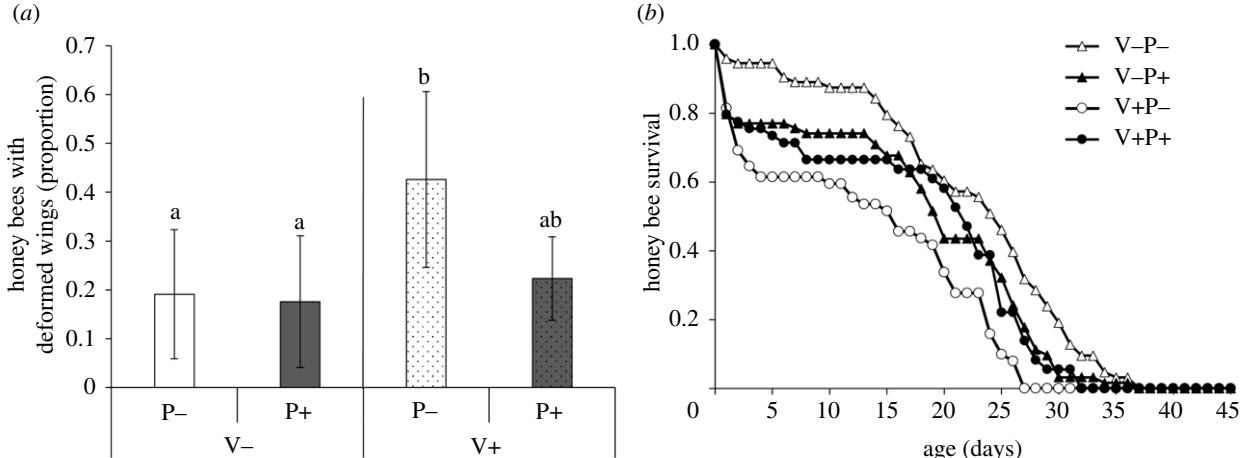

**Figure 2.** (*a*) Proportion of bees (mean ± s.d.) showing the characteristic symptoms of high viral infection levels which emerged from rearing cells treated or not with propolis (P+/P−) and infested or not with a mite (V+/V−). Bars marked with a different letter are significantly different at $p < 0.05$. (*b*) Survival of mite-infested (V+) and uninfested (V−) honeybees, from artificial rearing cells treated or not with propolis (P+/P−).

the survival of honeybees maintained both in treated and untreated cells (figure 2*b*; Cox regression analysis: H.R. = 1.852, $p < 0.001$), while the treatment of the rearing cells with propolis had no effect on bee survival (figure 2*b*; Cox regression analysis: H.R. = 1.063, $p = 0.656$). A significant interaction between propolis treatment and mite presence indicates that propolis in rearing cells significantly increases the survival of adult bees infested at the pupal stage (figure 2*b*; Cox regression analysis: H.R. = 0.372, $p < 0.001$) but not to the level of uninfested bees.

The phenotype of mite-infested bees emerging from cells treated with propolis is consistent with a lower DWV load in comparison with parasitized bees from untreated cells, similar to previous observations [16]. This suggests that, by impacting the survival and reproduction of mites, propolis may indirectly reduce the mite induced viral replication [22]. DWV is linked in a mutualistic symbiosis with the *Varroa* mite, which can promote its replication within the host, obtaining a facilitated feeding activity [23]. For this

reason, any effect on the mite can propagate to the virus, as we observed here where the presence of propolis inside the brood cells, affecting the survival and reproduction of parasitizing mites, resulted in a reduced percentage of symptomatic bees and increased survival. In sum, as a result of the effect of this natural pesticide on reproducing mites, the survival of honeybees emerging from propolis treated cells was significantly enhanced.

Since propolis is applied by bees to the honeycomb before oviposition, and the comb cells could be later used for the storage of honey and pollen, we assumed that—in view of its polarity—the honey stored in the cells could acquire some active compounds from propolis, as suggested by previous studies [3]. Therefore, to evaluate the possible effect of a propolis-enriched diet on the survival and viral load of adult bees parasitized by the mite during the pupal stage, adult bees emerging from untreated artificial cells containing or not a *Varroa* mite (V+ and V−, respectively) were maintained in cages under standardized environmental conditions and

*Proc. R. Soc. B* **288**: 20212101

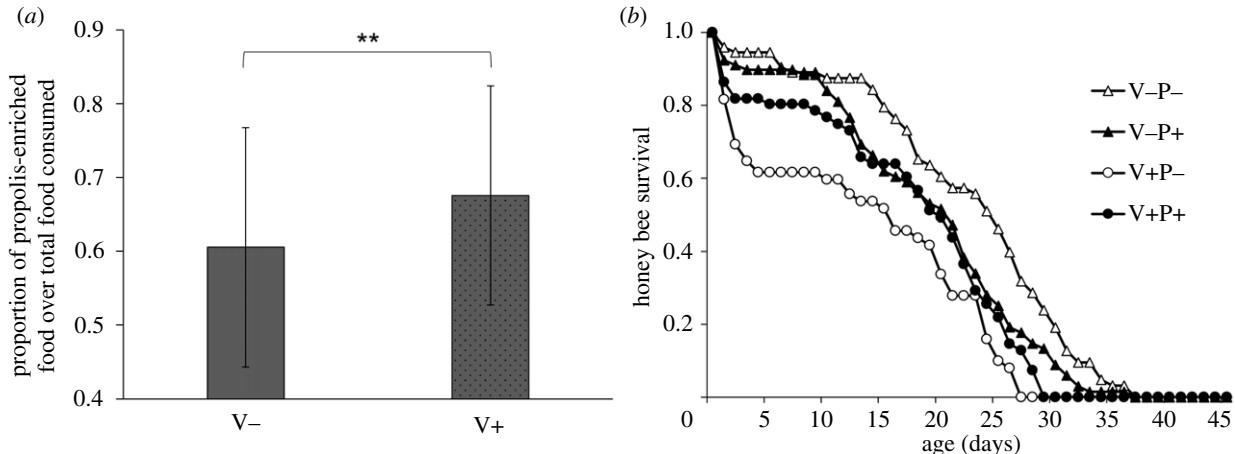

**Figure 3.** (*a*) Preference of mite-infested (V+) and uninfested bees (V−) for a propolis-enriched diet over a sugar only diet (mean ± s.d.). Two asterisks mark experimental groups differing at $p < 0.01$. Generalized linear mixed-effects models: $Z$-value = 2.852, $p = 0.004$. (*b*) Survival of adult bees that were infested with a mite or not at the pupal stage (V+/V−) and fed as adults with propolis or not (P+/P−).

fed ad libitum a sucrose solution complemented or not with a propolis extract (P+ and P−, respectively). The *Varroa*-infested group consumed proportionately more propolis diet than the *Varroa*-free group (figure 3*a*; V− versus V+; generalized linear mixed-effects models: $Z$-value = 2.852, $p = 0.004$). As expected, adult bees infested during the pupal stage had a significantly lower survival (figure 3*b*; Cox regression analysis: H.R. = 2.07, $p < 0.001$) than uninfested ones. In general, the addition of propolis to the diet did not affect the survival of adults (figure 3*b*; Cox regression analysis: H.R. = 1.06, $p = 0.655$), but propolis significantly decreased the mortality of mite-infested bees as highlighted by the significant mite × propolis interaction (figure 3*b*; Cox regression analysis: H.R. = 0.435, $p = 0.002$).

To check for any possible effect of a propolis-enriched diet on viral infection level, we tested the viral load of the bees used in this experiment and found no indication of a possible direct effect of propolis on the virus, neither on uninfested nor on mite-infested bees, which had a significantly higher viral load (electronic supplementary material, figure S3). To indirectly check for the presence of possible secondary infections triggered by mite infestation, we assessed the expression of two honeybee antimicrobial genes: Apidaecin and Defensin [24]. We found a significant upregulation of Apidaecin in mite-infested bees (electronic supplementary material, figure S4A); whereas Defensin upregulation only approached significance (electronic supplementary material, figure S4B). The relatively higher consumption of a propolis-enriched diet observed in mite-infested bees as compared to uninfested ones, suggests that bees can perceive some bioactive components from the food and preferentially feed on it. As a result of the propolis acquired through the diet, adult honeybees infested at the pupal stage survived longer. Apparently, longer survival is not related to a direct effect of propolis on viral infection; instead, a possible action against other microbial pathogens facilitated by the mite [25] appears to be more likely in view of our gene expression studies. Overall, this result clearly demonstrates the therapeutic activity of propolis on mite-infested bees, significantly deepening our understanding of the process of social medication in honeybees [7].

The subject of behavioural immunity in insects is receiving increasing attention since it was eventually appreciated that self-medication is not restricted to vertebrates, with their high cognitive abilities, but widespread among animals [26]. Here we show for the first time that, along with the already known mechanisms of behavioural immunity [27], honeybees can display a further level of defence based on the use of a substance to treat the environment where an ectoparasite reproduces. In this way a reduction of the survival and reproduction of the parasitic mite *V. destructor* inside the bee brood cells is achieved, with notable benefits for the bees developing in those cells. Furthermore, propolis can support mite-infested bees also after reaching the adult stage, probably by reducing possible secondary infections triggered by the mite.

Over time, the deposition behaviour of propolis into the hive has been negatively selected by beekeepers because this sticky material disturbs the handling of frames [4]. Our study, which is the first to deal with all the effects of propolis on *Varroa*, highlights the importance of this substance for colony health, suggesting that the development of strategies to stimulate resin collection and propolis storage into the hive could have a beneficial effect on bee health and should therefore be promoted. We hope that this work will stimulate further studies aiming at assessing the potential of propolis for the control of *V. destructor*: a strategic issue to preserve the sustainability of beekeeping and in turn food production.

## 3. Material and methods

### (a) Propolis collection

The propolis samples used in our bioassays were collected from 12 hives placed in the experimental apiary of the University of Sassari (latitude 40°46′23″ N, longitude 8°29′34″ E), using fine nylon mesh placed above the combs, in May 2018 (i.e. two months before the beginning of the experiments). Then, propolis samples were separated from the net, cleaned, by removing visible impurities, and stored in freezer at −18°C. To perform the experiments and chemical analysis, frozen samples were homogenized using a coffee mill (GS Arendo, Germany). To exclude the possible acaricide contamination of the propolis

used in our experiments, the samples were analysed by gas chromatography–ion trap-mass spectrometry (GC–ITMS analysis) following the method described by Pusceddu et al. [15].

## (b) Determination of the amount of resin in honeycomb

To evaluate the amount of resin inside the cells, honeycombs were extracted with different solvents. To this aim a piece of honeycomb weighting 3 g was placed in a 50 ml flask and subjected to three consecutive extractions with 10 ml of hexane, followed by extraction with $MeOH/H_2O$ (80/20, v/v). Extractions lasted 3 min in a vortex (Reax Top, Heidolph, Germany) and 30 min in a rotatory shaker. After centrifugation at 4000 r.p.m. at 10°C, the supernatants were collected, combined and evaporated under $N_2$. Using this method, both freshly built honeycombs and honeycombs ready for oviposition (as revealed by the presence of eggs within some cells) were analysed; the whole procedure was replicated six times.

## (c) Extraction of propolis

Ground propolis was weighted with an analytical balance and extracted three consecutive times with methanol/water solution (80/20, v/v) (1:5); after centrifugation for 12 min at 4000 r.p.m. and 10°C, the organic phases were combined. The resulting solution was evaporated under vacuum at room temperature, to obtain a paste, which was stored at −18°C before use.

## (d) Chromatographic analyses of propolis

Methanol and acetonitrile, analytical LC–MS grade, and ethanol 95% reagent grade were obtained from Sigma (Milan, Italy); 3, 4 dihydroxybenzoic acid, pirocathecol, vanillic acid, vanillin, 3,5 dihydroxycinnamaldeide, 2,5 dihydroxybenzoic acid, caffeic acid, paracumaric acid, ferulic acid, verbascoside, omovanillic acid, quercethin, narirutin, cinnamid acid, quercetin dihydrate, luteolin, naringenin, apigenin, kaempherol, galangin and syringic acid (I.S.) were purchased from Sigma-Aldrich (Milan, Italy). Stock standard solutions of single compounds were prepared in methanol ($\approx 1000$ mg $l^{-1}$), while mixed working standard solutions (100 µg $l^{-1}$) were prepared daily by dilution of the stock solutions with the eluent mixture. Calibration curves were prepared with five points using syringic acid as the internal standard. The Folin–Ciocalteau reagent was purchased from Sigma-Aldrich (St. Louis, MO, USA). HPLC water with conductivity lower than 18.2 MΩ was obtained using a Milli-Q system (Millipore, Bedford, MA, USA). Propolis, comb extracts and diet solution were analysed with an HPLC 1100 system, equipped with a DAD detector G1315A, an autosampler G1313A, a pump G1311A, a column oven G1316 (Agilent Technologies, Milan, Italy) and an integrated HP CHEMSTATION LC software. The following wavelengths: 280, 360 and 520 nm were monitored. The column was a Varian Polaris C18 (5 µm, 300 A, 250 mm × 4.6 mm). The solvents were phosphoric acid (0.22 M) (A) and acetonitrile/methanol (1/1, v/v) (B), with a flow of 1 ml $min^{-1}$. The gradient used for separation and analysis was $T = 0$, 96% A; $T = 40$, 50% A; $T = 45$, 40% A; $T = 60$, 0% A, hold for 10 min; column reconditioning was made at the initial conditions for 15 min. Quantitative analysis for each chemical was carried out using the proper calibration curve with the internal standard method (syringic acid at 150 mg $l^{-1}$). Total polyphenol content of propolis extract was determined by the Folin–Ciocalteu method, using a Cary 50 spectrophotometer (Agilent, Milan, Italy). For this purpose, 100 µl of the propolis solution or standard were added to 500 µl of Folin–Ciocalteau reagent and left to react for 5 min, thereafter 3 ml of 10% (w/v) sodium carbonate solution and ultrapure water, up to a final volume of 10 ml, were added. After incubation at room temperature for 90 min, the samples were read at $\lambda = 725$ nm against a blank using 1 cm path quartz cuvettes.

Quantitative analysis was carried out using the external standard method (gallic acid) by correlating absorbance (Abs) with concentration (400–8000 mg $kg^{-1}$) and expressing results in mg $kg^{-1}$ of gallic acid.

## (e) Preparation of propolis extract

One gram of propolis extract was dissolved in 2 ml of ethanol such that 10 µl of extract could be used to apply 5 mg of the resinous fraction to each rearing cell, roughly corresponding to the amount extracted from each honeycomb cell during the chemical analysis.

## (f) Preparation of propolis diet

One gram of propolis extract was dissolved in 1 ml of ethanol and then solubilized in 0.5 l of sucrose solution 50% (w/v), by stirring in a beaker with a magnetic anchor for 12 h. The composition of the diet that was used in the bioassays did not exactly replicate that of propolis extract, in that the water medium did not solubilize all the propolis paste leaving a resinous pellet which probably incorporates the missing compounds.

## (g) Effect of propolis in rearing cells on host and parasite fitness

Honeybee larvae and V. destructor adult females were sampled from hybrid A. m. ligustica × A. m. carnica colonies [28,29] located in the experimental apiary of the University of Udine, Dipartimento di Scienze AgroAlimentari, Ambientali e Animali (latitude 46°04′53.3″ N, longitude 13°12′33.1″ E). Bee larvae and mites were obtained from brood cells capped in the preceding 15 h, as described by Nazzi & Milani [18]. Fifth instar larvae were put into gelatin capsules (6.5 mm Ø; Agar Scientific Ltd.) with 1 (V+) or 0 (V−) mites and maintained in an incubator at 34.5°C, 75% R.H., dark, for 12 days [18]. In both cases, 15 h before the beginning of the experiment, artificial cells were treated with 10 µl of propolis extracts in ethanol (P+E+, treated) or ethanol only (P−E+ positive control) and kept under a hood, in order to let the ethanol to evaporate; the propolis extract was deposited inside the capsules using a pipette, then capsules were placed horizontally in a Petri dish and rotated by hand, to allow the extract to spread over the internal surface of the cell. Another group of cells was left untreated (P−E−, negative control). The following experimental groups were considered:

— uninfested larvae in capsules treated with propolis extracts in ethanol (V−P+E+),
— uninfested larvae in untreated capsules (V−P−E−),
— uninfested larvae in capsules treated with ethanol only (V−P−E+),
— mite-infested larvae in capsules treated with propolis extracts in ethanol (V+P+E+),
— mite-infested larvae in untreated capsules (V+P−E−),
— mite-infested larvae in capsules treated with ethanol only (V+P−E+).

After 12 days, the following parameters were checked: mortality and weight of emerging bees, mortality of mites and proportion of reproducing mites (i.e. fertility = reproducing mites/surviving foundress mites). The weight of the emerging adults, parasitized during the pupal stage, was considered only if the mite had survived until the emergence. Mite-infested bees and controls that emerged from untreated cells and from cells treated with propolis were transferred into plastic cages (185 × 105 × 85 mm), maintained in a climatic chamber (34.5°C, 75% R.H., dark) and fed with sugar candy (68% sucrose, 16% glucose, 16% fructose) ad libitum. Cages were checked daily to count and remove the dead bees for survival statistics. The

experiment was replicated three times, using 80–90 larvae per treatment and about 24 adult bees per group, in each of the three replicates. Survival rates of uninfested and mite-infested adult bees emerging from cells treated with propolis or not were compared by means of Cox proportional hazard regression analysis using R statistical software, v. 3.6.2. [30]. The proportion of bees with crippled wings at emergence were compared with the Mantel–Haenszel test, and the results corrected for multiple comparisons according to Benjamini and Hochberg. The state of wings of the bees belonging to each experimental group was visually assessed both at the emergence and then confirmed when removing the dead bees from the rearing cages, since this phenotype is not easily noted at the emergence, when wings are not always fully extended. Weight and mortality of mite-infested or uninfested bees emerging from cells treated or not with propolis were compared with two-way ANOVA test; in the case of bee mortality, the angular transformation was applied to data before testing. Comparisons of mite mortality and fertility in cells treated or not with propolis were conducted using Mantel–Haenszel test. For this analysis, 40–56 mites per group were used in each of the three replicates. Statistical analyses were performed using R statistical software, v. 3.3.2 [31]. All data and the details of the statistical analyses are reported in electronic supplementary material, dataset S1.

## (h) Effect of a propolis-enriched diet on adult survival

Mite-infested and uninfested newly emerged bees collected from the untreated cells were transferred into plastic cages as described above. Bees were fed ad libitum two different diets: sucrose water solution (P−) or sucrose water solution supplemented with propolis extracts (P+). Cages were checked daily to count and remove the dead bees. Seven days after the eclosion, three bees per treatment were sampled in liquid nitrogen and stored at −80°C for subsequent molecular analysis. Comparisons of survival rates of the uninfested and mite-infested bees fed different diets were conducted by means of Cox proportional hazard regression analysis in R statistical software, v. 3.6.2. [30]. Comparison of propolis consumption in uninfested and mite-infested bees was conducted using a generalized linear mixed-effects model analysis. Measurement of food intake were made daily for 19 days after emergence. For this analysis, about 24 bees per group were used in each of the three replicates. All data and the details of the statistical analyses are reported in electronic supplementary material, dataset S1.

## (i) Analysis of deformed wing virus, Apidaecin and Defensin

The whole body of sampled bees was homogenized using mortar and pestle in liquid nitrogen. Total RNA was extracted from each bee according to the procedure provided with the RNeasy Plus mini kit (Qiagen, Germany). The amount of RNA in each sample was quantified with a NanoDrop spectrophotometer (ThermoFisher, US). cDNA was synthetized starting from 500 ng of RNA following the manufacturer specifications (PROMEGA, Italy). Additional negative control samples containing no RT enzyme were included. DWV, Apidaecin and Defensin relative quantification across the treatments was obtained by qRT-PCR using the following primers: DWV (F: GGTAAGC-GATGGTTGTTTG, R: CCGTGAATATAGTGTGAGG [32]), Apidaecin (F: TTTTGCCTTAGCAATTCTTGTTG, R: GAAGGTC-GAGTAGGCGGATCT [24]), Defensin (F: CATGGCTAAT GCCGGAGAGG, R: CTGCACCAGCTTGAAGAGC [24]) and β-actin (F: GATTTGTATGCCAACACTGTCCTT, R: TTGCATTC-TATCTGCGATTCCA [23]). Ten nanograms of cDNA from each sample were analysed using SYBRgreen dye (Ambion) according to the manufacturer specifications, on a BioRad CFX96 Touch Real

time PCR Detector. The following thermal cycling profiles were adopted: one cycle at 95°C for 10 min, 40 cycles at 95°C for 15 s and 60°C for 1 min, and one cycle at 68°C for 7 min. Relative viral quantities were analysed adopting with the $2^{-\Delta\Delta Ct}$ method [33] using β-actin as housekeeping gene. Normalized values were analysed using two-way ANOVA test. In total, six to nine bees per treatment were analysed. All data and the details of the statistical analyses are reported in electronic supplementary material, dataset S1.

## (j) Diet choice bioassay

Honeybee workers were sampled from six colonies of A. m. ligustica of the experimental apiary of the Dipartimento di Agraria of the University of Sassari located in northwestern Sardinia (latitude 40°46′23″, longitude 8°29′34″). To obtain Varroa-infested emerging bees, we collected combs with brood ready to emerge from three Varroa-infested colonies. These combs were kept under observation in the laboratory for 8 h. Each emerging bee was checked for the presence or absence of the mite inside the cell. In some cases, the bees were helped to emerge by removing the cell cap. A similar procedure was used to obtain uninfested bees from three Varroa-free colonies. Subsequently, the bees were placed in metal cages (100 × 100 × 50 mm). Each cage had a group of 30 infested or uninfested honeybees (composed of 10 bees from each of the three colonies), so as to exclude any colony genotypic effect. The cages were kept in an incubator (31°C, 70% R.H., dark) and two types of diet were supplied ad libitum in each cage, as follows: 50% (w/v) sucrose solution (P−) and sucrose solution supplemented with propolis (P+). The experiment was replicated three times with three independent cages per replicate. During the experiment, cages were checked daily to count and remove dead bees. The amount of diet consumed (with and without propolis) in each cage was measured by daily weighing the two syringes containing the diet solutions. The experiment finished whenever in at least one cage of a replicate the number of bees was less than 5 to reduce the error in the assessment of food consumption by weight per bee. Comparisons of the relative proportion of propolis diet consumption between uninfested and infested bees were conducted with a generalized linear mixed model (GLMM) analysis with a binomial distribution error. The experiment-time and cages were considered as random factors. Statistical analysis was performed using R statistical software, v. 3.3.2. [31]. All data and the details of the statistical analyses are reported in electronic supplementary material, dataset S1.

Data accessibility. The data are provided in electronic supplementary material [34].

Authors' contributions. M.P.: conceptualization, data curation, investigation, writing—original draft; D.A.: conceptualization, data curation, investigation; I.F.: conceptualization; D.F.: data curation, investigation; V.Z.: conceptualization, investigation; A.A.: investigation; A.S.: conceptualization, data curation, funding acquisition, investigation, supervision, writing—original draft, writing—review and editing; F.N.: conceptualization, data curation, funding acquisition, investigation, supervision, writing—original draft, writing—review and editing.

All authors gave final approval for publication and agreed to be held accountable for the work performed therein.

Competing interests. Authors declare no competing interests.

Funding. This research was funded by: Fondazione di Sardegna—year 2016 and by Regione Autonoma della Sardegna (Italy), L.R. 7/2007—year 2016, project: 'Self-medication in the hive: propolis and venom against the honeybee ectoparasite Varroa destructor'; Università degli Studi di Sassari 'Fondo di Ateneo per la ricerca'; the Italian Ministry of University, PRIN 2017—UNICO (2017954WNT); Regione Autonoma del Friuli Venezia Giulia (Italy), L.R. 6/2010, Laboratorio Apistico Regionale, 2020–2022.

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
