## [Peer Review File · Proceedings of the Royal Society B: Biological Sciences]

Review History

RSPB-2021-0466.R0 (Original submission)

Review form: Reviewer 1

Recommendation

Major revision is needed (please make suggestions in comments)

Scientific importance: Is the manuscript an original and important contribution to its field?

Good

General interest: Is the paper of sufficient general interest?

Good

Quality of the paper: Is the overall quality of the paper suitable?

Marginal

Is the length of the paper justified?

Yes

Should the paper be seen by a specialist statistical reviewer?

No

Do you have any concerns about statistical analyses in this paper? If so, please specify them explicitly in your report.

No

It is a condition of publication that authors make their supporting data, code and materials available - either as supplementary material or hosted in an external repository. Please rate, if applicable, the supporting data on the following criteria.

Is it accessible?

Yes

Is it clear?

Yes

Is it adequate?

No

Do you have any ethical concerns with this paper?

No

Comments to the Author

I was very excited to read this article, but after some reflection feel that the data are not as strong as the authors claim.

First, it is good to know that newly built wax combs, prior to oviposition, contain signals of resin/ propolis (phenol content). However, it is unclear whether bees are intentionally mixing resin into the wax, or if their tarsal "foot" prints inadvertently mix resin into the combs.

The second, potentially important finding is that the presence of propolis in cells can reduce mite fertility. However, they do not present any methods or data on this reduced fertility. How did they define fertility? It should be by the number of offspring produced per foundress mite. These data must be presented because they are critical to understanding the extent that propolis is actually a "pesticide" against Varroa.

In the experiment the authors applied the propolis powder to gelatin capsules, not to wax cells, and reared the pupae in the laboratory. Mites were introduced, or not, into the capsules along with 5th instar larvae. Thus, the larvae and pupae and mites were in direct contact with the propolis. What would happen if the gelatin capsules also contained beeswax? I think the authors need to combine the propolis powder with beeswax to determine if they still see the same effect on mite fertility and adult bee survivorship. I would be much more convinced of the data (along with seeing data on mite fertility).

The effects of the propolis on viral infection and adult survivorship are very interesting and worthy of continued study.

In summary, I am not fully convinced that propolis within wax combs affects the fertility and survivorship of Varroa, simply because Varroa is a problem for individual bee survivorship and colony survivorship worldwide, and yet *Apis mellifera* collects propolis everywhere it lives, despite selection against propolis-collecting strains of bees by beekeepers. At some dose propolis must be toxic to bee larvae. I'm afraid this article will stimulate beekeepers to paint or dip their combs into propolis extracts, where it could cause some problems. The authors need to add some caveats to warn against doing this.

Review form: Reviewer 2

Recommendation

Accept with minor revision (please list in comments)

Scientific importance: Is the manuscript an original and important contribution to its field?

Excellent

General interest: Is the paper of sufficient general interest?

Good

Quality of the paper: Is the overall quality of the paper suitable?

Good

Is the length of the paper justified?

Yes

Should the paper be seen by a specialist statistical reviewer?

No

Do you have any concerns about statistical analyses in this paper? If so, please specify them explicitly in your report.

No

It is a condition of publication that authors make their supporting data, code and materials available - either as supplementary material or hosted in an external repository. Please rate, if applicable, the supporting data on the following criteria.

Is it accessible?

No

Is it clear?

Yes

Is it adequate?

Yes

Do you have any ethical concerns with this paper?

No

Comments to the Author

The authors report on the anti-Varroa properties of propolis incorporated wax cells. This work could be of great practical benefit for beekeepers seeking to control this deadly parasite while also providing good evidence that propolis is incorporated into wax -- a controversial topic in the bee research community. Overall I found the manuscript to be well-written, but I think several areas could be clarified to make it more useful for the research community.

Major points:

Please provide the phenolic composition of wax, as summarized in Table 1, for honeycomb. The phenolic constituents of propolis are reported in Table S1, but a similar table needs to be added for the phenolics found in comb.

The immature bees in this study were reared in gelatin capsules, but that is not clear in the way that the treatments are reported in lines 251-256. Please change "cells" to "capsules".

A larger issue with the treatment of gelatin capsules with propolis extract is the fact that gelatin has very different chemical properties than wax. It should be mentioned as a caveat in the discussion that bees may be exposed differently when propolis is incorporated into highly lipophilic wax vs. gelatin. I suspect the phenolics are much more bioavailable when coated on gelatin -- it would have been good to see this addressed in the experiments, but at a minimum it should be addressed in the discussion.

At several points through the manuscript the term "sprinkled" is used to describe application of propolis to the capsules (e.g. line 247). This term comes across as overly casual. Please be more descriptive about the application method at line 247 and replace "sprinkled" with "applied" or another verb in the legend for Figure 1.

The paragraph starting on line 166 should be revised to improve readability and to avoid implying that *A. mellifera* has evolved over the last 100+ years to use propolis against varroa mites. The sentence at lines 170-172, is especially problematic in this regard.

In the methods on line 266 it states that "crippled wings" were observed at emergence. How was the "crippled" status of emerging bees determined. Was this purely visual? Provide more details on how this was determined by observers.

Minor points:

Line 44: Remove word "dramatically"

Lines 73-74: Use consistent units -- either g/kg or mg/g for both

Line 75: "to reinforce the structure and disinfect the cells" should probably be removed here. Or references should be provided to support this statement.

Line 100: "narcoleptic" is probably not the right word in this context.

Line 108: change "had _not_ effect" to "had _no_ effect"

Line 120: Change "indicates" to "is consistent with"

Figure 3: Remove "(adult)" from Fig. 3A

Line 159: "indicates" is too strong a term as more research is needed. Change to "suggests"

Line 261: Provide the composition of the sugar candy used

Line 288: Check spelling of spectrophotometer

Line 205: Add "the" to "observation in _the_ laboratory"

Decision letter (RSPB-2021-0466.R0)

26-Apr-2021

Dear Dr Nazzi:

I am writing to inform you that your manuscript RSPB-2021-0466 entitled "Honey bees use propolis as a natural pesticide against their major ectoparasite" has, in its current form, been rejected for publication in Proceedings B.

This action has been taken on the advice of referees, who have recommended that substantial revisions are necessary. With this in mind we would be happy to consider a resubmission, provided the comments of the referees are fully addressed. However please note that this is not a provisional acceptance.

Yours sincerely,
 Professor Loeske Kruuk
 Editor
 mailto: proceedingsb@royalsociety.org

Associate Editor
 Board Member: 1
 Comments to Author:

While the topic of this study has the potential to be of great general interest as well as practical interest for beekeepers, the study is lacking some key data and both referees are asking for additional data that may require extra experimentation. Referee 1 points out that data on mite fertility are not provided, this needs to be remedied in a revision. Referee 2 is also asking for data on phenolics in comb (not just propolis). Referee 2 also finds the data to be not accessible. Finally, referee 1 asks that the authors temper their conclusions, and both referees had multiple additional major and minor concerns with the study. Thus, a substantial revision would be required to make the study publishable in PRSB.

Reviewer(s)' Comments to Author:

Referee: 1

Comments to the Author(s)

I was very excited to read this article, but after some reflection feel that the data are not as strong as the authors claim.

First, it is good to know that newly built wax combs, prior to oviposition, contain signals of resin/ propolis (phenol content). However, it is unclear whether bees are intentionally mixing resin into the wax, or if their tarsal "foot" prints inadvertently mix resin into the combs.

The second, potentially important finding is that the presence of propolis in cells can reduce mite fertility. However, they do not present any methods or data on this reduced fertility. How did they define fertility? It should be by the number of offspring produced per foundress mite.

These data must be presented because they are critical to understanding the extent that propolis is actually a "pesticide" against Varroa.

In the experiment the authors applied the propolis powder to gelatin capsules, not to wax cells, and reared the pupae in the laboratory. Mites were introduced, or not, into the capsules along with 5th instar larvae. Thus, the larvae and pupae and mites were in direct contact with the propolis. What would happen if the gelatin capsules also contained beeswax? I think the authors need to combine the propolis powder with beeswax to determine if they still see the same effect on mite fertility and adult bee survivorship. I would be much more convinced of the data (along with seeing data on mite fertility).

The effects of the propolis on viral infection and adult survivorship are very interesting and worthy of continued study.

In summary, I am not fully convinced that propolis within wax combs affects the fertility and survivorship of *Varroa*, simply because *Varroa* is a problem for individual bee survivorship and colony survivorship worldwide, and yet *Apis mellifera* collects propolis everywhere it lives, despite selection against propolis-collecting strains of bees by beekeepers. At some dose propolis must be toxic to bee larvae. I'm afraid this article will stimulate beekeepers to paint or dip their combs into propolis extracts, where it could cause some problems. The authors need to add some caveats to warn against doing this.

Referee: 2

Comments to the Author(s)

The authors report on the anti-*Varroa* properties of propolis incorporated wax cells. This work could be of great practical benefit for beekeepers seeking to control this deadly parasite while also providing good evidence that propolis is incorporated into wax -- a controversial topic in the bee research community. Overall I found the manuscript to be well-written, but I think several areas could be clarified to make it more useful for the research community.

Major points:

Please provide the phenolic composition of wax, as summarized in Table 1, for honeycomb. The phenolic constituents of propolis are reported in Table S1, but a similar table needs to be added for the phenolics found in comb.

The immature bees in this study were reared in gelatin capsules, but that is not clear in the way that the treatments are reported in lines 251-256. Please change "cells" to "capsules".

A larger issue with the treatment of gelatin capsules with propolis extract is the fact that gelatin has very different chemical properties than wax. It should be mentioned as a caveat in the discussion that bees may be exposed differently when propolis is incorporated into highly lipophilic wax vs. gelatin. I suspect the phenolics are much more bioavailable when coated on gelatin -- it would have been good to see this addressed in the experiments, but at a minimum it should be addressed in the discussion.

At several points through the manuscript the term "sprinkled" is used to describe application of propolis to the capsules (e.g. line 247). This term comes across as overly casual. Please be more descriptive about the application method at line 247 and replace "sprinkled" with "applied" or another verb in the legend for Figure 1.

The paragraph starting on line 166 should be revised to improve readability and to avoid implying that *A. mellifera* has evolved over the last 100+ years to use propolis against *Varroa* mites. The sentence at lines 170-172, is especially problematic in this regard.

In the methods on line 266 it states that "crippled wings" were observed at emergence. How was the "crippled" status of emerging bees determined. Was this purely visual? Provide more details on how this was determined by observers.

Minor points:

Line 44: Remove word "dramatically"

Lines 73-74: Use consistent units -- either g/kg or mg/g for both

Line 75: "to reinforce the structure and disinfect the cells" should probably be removed here. Or references should be provided to support this statement.

Line 100: "narcoleptic" is probably not the right word in this context.

Line 108: change "had _not_ effect" to "had _no_ effect"

Line 120: Change "indicates" to "is consistent with"

Figure 3: Remove "(adult)" from Fig. 3A

Line 159: "indicates" is too strong a term as more research is needed. Change to "suggests"

Line 261: Provide the composition of the sugar candy used

Line 288: Check spelling of spectrophotometer

Line 205: Add "the" to "observation in _the_ laboratory"

Author's Response to Decision Letter for (RSPB-2021-0466.R0)

See Appendix A.

RSPB-2021-2101.R0

Review form: Reviewer 1

Recommendation

Accept with minor revision (please list in comments)

Scientific importance: Is the manuscript an original and important contribution to its field?

Good

General interest: Is the paper of sufficient general interest?

Good

Quality of the paper: Is the overall quality of the paper suitable?

Good

Is the length of the paper justified?

Yes

Should the paper be seen by a specialist statistical reviewer?

No

Do you have any concerns about statistical analyses in this paper? If so, please specify them explicitly in your report.

No

It is a condition of publication that authors make their supporting data, code and materials available - either as supplementary material or hosted in an external repository. Please rate, if applicable, the supporting data on the following criteria.

Is it accessible?

Yes

Is it clear?

Yes

Is it adequate?

Yes

Do you have any ethical concerns with this paper?

No

Comments to the Author

In general, the authors have greatly improved the presentation of the data in this manuscript. The results are exciting, but I think it is important to further temper the wording and conclusions even more. The abstract is fine, but for example, line 175 of the conclusion states, "... a significant reduction of the survival and reproduction of .. the mites in brood cells..." While it is true there was a statistically significant reduction in survival and reproduction of the mites, in actuality there was about an 18% reduction in mite survival, and of the mites that survived, 44% were infertile, or less than 8% overall. Relative to mite treatments, or resistance behaviors, these reductions are relatively small (but still exciting!). My suggestions for tempering the wording are as follows:

1. Line 175: We found an 18% (or whatever actual mean was) reduction in the survival of the mites, and 8% of the surviving mites were infertile, indicating that some reduction in survival and reproduction of mites in brood cells is achieved, with benefits for the bees developing in those cells."
2. The reduction in DWV signs when bees developed in gelatin capsules (Fig 2A), relative to DWV in adult bees fed sugar solution with propolis (Fig S3) is interesting, and hints at the possible mode of action of propolis on virus, which is evidently by volatiles or contact and not by oral inoculation. I think the authors could draw out this comparison in the Discussion, particularly in lines 127-133 and line 158 (which without more explanation to line 158, may appear to contradict lines 127-133 to a casual reader).
3. Line 108: "... propolis does not seem to be able to counteract the negative, direct effects of mite infestation on developing bees.." relative to ?? need to complete sentence.
4. Explanation of the survival curves (Figs 2B and 3B): Lines 115 ("Propolis increases survival of adult bees" and 146: "... but propolis significantly decreased the mortality of mite infested bees..."), please consider adding, "but not to the level of uninfested bees."
5. line 127: "propolis may reduce the indirect effects... and in particular, may reduce the activation of viral replication."
6. line 31: *Ascospaera apis* is the causative agent of chalkbrood, not stonebrood.

some minor English grammar suggestions

line 82: different from the wax that comprises the brood cells, but does not differ from the silk cocoon layer...

Decision letter (RSPB-2021-2101.R0)

18-Oct-2021

Dear Dr Nazzi,

I am pleased to inform you that your manuscript RSPB-2021-2101 entitled "Honey bees use propolis as a natural pesticide against their major ectoparasite" has been accepted for publication in Proceedings B.

The revised manuscript was reviewed by one of the original referees, who has recommended publication, but also suggest some further minor, but still important, revisions to your manuscript. Therefore, I invite you to respond to the referee's comments and revise your manuscript. Please pay particular attention to the need to be clear about the magnitude of the effects you are seeing, in terms of the effect sizes rather than just their significance: as the referee states, the management value of your results makes it especially important to be clear about this. Because the schedule for publication is very tight, it is a condition of publication that you submit the revised version of your manuscript within 7 days. If you do not think you will be able to meet this date please let us know.

Sincerely,

Professor Loeske Kruuk

Reviewer(s)' Comments to Author:

Referee: 1

Comments to the Author(s).

In general, the authors have greatly improved the presentation of the data in this manuscript.

The results are exciting, but I think it is important to further temper the wording and conclusions even more. The abstract is fine, but for example, line 175 of the conclusion states, "... a significant reduction of the survival and reproduction of .. the mites in brood cells..." While it is true there was a statistically significant reduction in survival and reproduction of the mites, in actuality there was about an 18% reduction in mite survival, and of the mites that survived, 44% were infertile, or less than 8% overall. Relative to mite treatments, or resistance behaviors, these reductions are relatively small (but still exciting!). My suggestions for tempering the wording are as follows:

1. Line 175: We found an 18% (or whatever actual mean was) reduction in the survival of the mites, and 8% of the surviving mites were infertile, indicating that some reduction in survival

and reproduction of mites in brood cells is achieved, with benefits for the bees developing in those cells.”

2. The reduction in DWV signs when bees developed in gelatin capsules (Fig 2A), relative to DWV in adult bees fed sugar solution with propolis (Fig S3) is interesting, and hints at the possible mode of action of propolis on virus, which is evidently by volatiles or contact and not by oral inoculation. I think the authors could draw out this comparison in the Discussion, particularly in lines 127-133 and line 158 (which without more explanation to line 158, may appear to contradict lines 127-133 to a casual reader).

3. Line 108: “... propolis does not seem to be able to counteract the negative, direct effects of mite infestation on developing bees..” relative to ?? need to complete sentence.

4. Explanation of the survival curves (Figs 2B and 3B): Lines 115 (“Propolis increases survival of adult bees” and 146: (“... but propolis significantly decreased the mortality of mite infested bees...”), please consider adding, “but not to the level of uninfested bees.”

5. line 127: “propolis may reduce the indirect effects... and in particular, may reduce the activation of viral replication.”

6. line 31: *Ascospaera apis* is the causative agent of chalkbrood, not stonebrood.

some minor English grammar suggestions

line 82: different from the wax that comprises the brood cells, but does not differ from the silk cocoon layer...

Author's Response to Decision Letter for (RSPB-2021-2101.R0)

See Appendix B.

Decision letter (RSPB-2021-2101.R1)

22-Nov-2021

Dear Dr Nazzi

I am pleased to inform you that your manuscript entitled "Honey bees use propolis as a natural pesticide against their major ectoparasite" has been accepted for publication in Proceedings B.

Your article has been estimated as being 8 pages long. Our Production Office will be able to confirm the exact length at proof stage.

Data Accessibility section

Open Access

Paper charges

Sincerely,

Proceedings B

Appendix A

Associate Editor

Board Member: 1

Comments to Author:

While the topic of this study has the potential to be of great general interest as well as practical interest for beekeepers, the study is lacking some key data and both referees are asking for additional data that may require extra experimentation. Referee 1 points out that data on mite fertility are not provided, this needs to be remedied in a revision. Referee 2 is also asking for data on phenolics in comb (not just propolis). Referee 2 also finds the data to be not accessible. Finally, referee 1 asks that the authors temper their conclusions, and both referees had multiple additional major and minor concerns with the study. Thus, a substantial revision would be required to make the study publishable in PRSB.

Dear Editor,

we carefully revised the manuscript keeping into account all the observations from the referees. In particular, we took the actions detailed below.

We thank you for pointing to our attention the weakness points of our work whose quality is now certainly improved.

Referee 1 points out that data on mite fertility are not provided, this needs to be remedied in a revision.

Actually, we provided mite fertility data (please see fig. 1B and lines 77-83 in Main text_NEW_CLEAN file; please consider also that we attached an excel file with all the raw data that were used to prepare the paper's figures).

We believe that the problem highlighted by referee 1 may have arisen from the fact that we referred to the standard definition of fertility (i.e. proportion of reproducing mites, see: Dieteman et al., Standard methods on Varroa research. Journal of Apicultural Research, 52 (2013)), whereas reviewer 1 refers to another definition of this important demographic parameter (i.e. number of offspring per foundress mites that was not reported in our manuscript).

We have not fixed the problem by providing an unequivocal definition of fertility in the methods and highlighting the parameter we used in the text (please, see lines 88-91, 96-99, 259-260 in Main text_NEW_CLEAN file).

Referee 2 is also asking for data on phenolics in comb (not just propolis).

We have now added those data as a further column in table S1.

We also slightly changed the wording in the results section (please, see lines 70-76 in Supplementary Information_NEW_CLEAN file).

Referee 2 also finds the data to be not accessible.

As we normally do, we added an excel file including all the data related to the experimental work; there anybody can find both the raw data, the resulting graphs and the statistical analysis.

Finally, referee 1 asks that the authors temper their conclusions

We moderated our conclusions according to the referee's request (please, see lines 175-178 and 183-185 in Main text_NEW_CLEAN file).

and both referees had multiple additional major and minor concerns with the study.

We reviewed the manuscript throughout according to both reviewers' suggestions.

Reviewer(s)' Comments to Author:

Referee: 1

Comments to the Author(s)

I was very excited to read this article, but after some reflection feel that the data are not as strong as the authors claim.

We are glad that referee 1 share our interest for this important subject; below we try to show that our results are strong enough to support our conclusions.

First, it is good to know that newly built wax combs, prior to oviposition, contain signals of resin/propolis (phenol content). However, it is unclear whether bees are intentionally mixing resin into the wax, or if their tarsal "foot" prints inadvertently mix resin into the combs.

Although it is certainly possible that some of the propolis that can be extracted from the honeycomb results from unintentional contamination, we do not think that the amount of propolis we recovered in the honey combs (up to 5% of phenol rich fraction, see table 1 and S1) is consistent with this origin.

The second, potentially important finding is that the presence of propolis in cells can reduce mite fertility. However, they do not present any methods or data on this reduced fertility. How did they define fertility? It should be by the number of offspring produced per foundress mite. These data must be presented because they are critical to understanding the extent that propolis is actually a "pesticide" against Varroa.

To us, in agreement with standard practice among bee parasitologists (see: Dieteman et al., Standard methods on Varroa research. Journal of Apicultural Research, 52 (2013)), "fertility" is the proportion of reproducing mites, that is the ratio between the number of mites producing at least one offspring and the total number of mites used in an experiment that survived until bee eclosion. Instead, the number of offspring produced per fertile mite is normally defined as "fecundity"; whereas, in some cases, a further parameter can be used that is the number of offspring per foundress mite.

There are some good reasons to consider the proportion of reproducing mites rather than the number of offspring per reproducing mites or the number of offspring per foundress mites.

Firstly, since only about 1 offspring (1.1 on average) out of the total number produced by a fertile mite in a brood cell can survive after bee eclosion (see Martin's old studies), fecundity has a much more limited impact on the dynamics of mite infestation than fertility defined as above and is therefore rarely considered. Secondly, by studying the distribution of mites with a given number of offspring, it has been show that mite reproduction needs to be triggered only once (see Di Prisco et al., A mutualistic symbiosis between a parasitic mite and a pathogenic virus undermines honey bee immunity and health. PNAS, 113: 3203-3208 (2016)); therefore, fertility defined as above, better captures the possible impact of any factor on mite reproduction. Lastly, assessing fertility is much easier and precise than counting each offspring since unfertile eggs or dead protonymphs can easily escape even the more careful observation.

For these reasons, in our study, we considered the proportion of reproducing mites and assessed the effect of propolis on this parameter which has got some clear implications for the parasite dynamics inside the hive. However, thanks to referee 1, we have now realized that we wrongly assumed that this definition is universal and did not define fertility in the methods.

We believe that this misunderstanding is at the base of the perplexity of referee 1. We hope that our results look now more convincing.

We now provide a clear definition of fertility both in the methods and in the results section where we speak about this (please, see lines 87-91, 96-99, 259-260 in Main text_NEW_CLEAN file).

In the experiment the authors applied the propolis powder to gelatin capsules, not to wax cells, and reared the pupae in the laboratory. Mites were introduced, or not, into the capsules along with 5th instar larvae. Thus, the larvae and pupae and mites were in direct contact with the propolis. What would happen if the gelatin capsules also contained beeswax? I think the authors need to combine the propolis powder with beeswax to determine if they still see the same effect on mite fertility and adult bee survivorship. I would be much more convinced of the data (along with seeing data on mite fertility).

The reason why we used gelatin capsules for testing any possible effect of propolis on mite reproduction is that the mites do not reproduce inside artificial cells made of wax (see Nazzi and Milani, A technique for reproduction of *Varroa jacobsoni* Oud under laboratory conditions. *Apidologie* 25: 579-584 (1994)). Instead, to our knowledge, gelatin capsules are the only artificial cells where invading mites from just capped brood cells can reproduce. Clearly, if reproduction has already been triggered, as in the case of mites from brood cells capped by more than 24 hours, parasites would reproduce anywhere but now mites would not be homogeneous as far as their physiological conditions are concerned.

However, it is worth noting that one of the most important differences between natural brood cells (where mite reproduction takes place) and artificial wax cells (where mite reproduction is rare at most) is the cocoon spun by larvae inside the cells. Under the chemical point of view, the silk made cocoon can be regarded as a polar layer internally lining the wax brood cell, which is strictly apolar. Under this respect, gelatine nicely mimics this internal polar layer that can be found inside natural brood cells, whereas wax would not and this may have important consequences for the fate of the polar phenolic compounds from propolis. In fact, the phenolic compounds from propolis inside a wax cells would likely float on the surface of the apolar wax and would be much more available to the mite; in this case any possible biological effect of propolis could be attributed to this peculiar situation that is relatively far from natural.

Anyway, this is an important point which we did not comment in the previous version of the manuscript; therefore, we have now included a short comment about this (please, see lines 80-83 in Main text_NEW_CLEAN file).

The effects of the propolis on viral infection and adult survivorship are very interesting and worthy of continued study.

We thank referee 1 for noting this; we'll certainly continue along this route of enquiry.

*In summary, I am not fully convinced that propolis within wax combs affects the fertility and survivorship of *Varroa*, simply because *Varroa* is a problem for individual bee survivorship and colony survivorship worldwide, and yet *Apis mellifera* collects propolis everywhere it lives, despite selection against propolis-collecting strains of bees by beekeepers. At some dose propolis must be toxic to bee larvae. I'm afraid this article will stimulate beekeepers to paint or dip their combs into propolis extracts, where it could cause some problems. The authors need to add some caveats to warn against doing this.*

We hope that the explanations provided above to better clarify our results convinced referee 1 about the effect of propolis on the survival and reproduction of the *Varroa* mite. Moreover, we hope that the new version of the manuscript looks more readable and free of omissions preventing a clear understanding of the results and their implications.

We like her/his remark about the fact that despite propolis collection by bees, the mite still represent a problem worldwide. Indeed, we think that tolerance against the mite involves multiple mechanisms (e.g. hygienic behavior, reduced mite fertility, grooming, etc.) and not just increased propolis collection.

Moreover, we fully agree that the information we are presenting here can be misunderstood by beekeepers and may stimulate them to paint their combs with propolis extract, etc.. We did our best not to overemphasize in the new text the immediate practical implications of the scientific data we present here, that certainly need extensive field experiments before leading to any possible practical application by beekeepers (please, see lines 183-185 in Main text_NEW_CLEAN file).

Referee: 2

Comments to the Author(s)

The authors report on the anti-Varroa properties of propolis incorporated wax cells. This work could be of great practical benefit for beekeepers seeking to control this deadly parasite while also providing good evidence that propolis is incorporated into wax -- a controversial topic in the bee research community. Overall I found the manuscript to be well-written, but I think several areas could be clarified to make it more useful for the research community.

We thank referee 2 for her/his positive evaluation of our manuscript which has now been further improved, trying to clarify all the aspects that needed attention.

Major points:

Please provide the phenolic composition of wax, as summarized in Table 1, for honeycomb. The phenolic constituents of propolis are reported in Table S1, but a similar table needs to be added for the phenolics found in comb.

We have now added those data as a further column in table S1 (please, see Supplementary Information_NEW_CLEAN file).

We also slightly changed the wording in the results section (please, see lines 70-76 in Main text_NEW_CLEAN file).

The immature bees in this study were reared in gelatin capsules, but that is not clear in the way that the treatments are reported in lines 251-256. Please change "cells" to "capsules".

Thank you for pointing to our attention that potential source of misunderstanding. The text was changed according to the suggestion.

A larger issue with the treatment of gelatin capsules with propolis extract is the fact that gelatin has very different chemical properties than wax. It should be mentioned as a caveat in the discussion that bees may be exposed differently when propolis is incorporated into highly lipophilic wax vs. gelatin. I suspect the phenolics are much more bioavailable when coated on gelatin -- it would have been good to see this addressed in the experiments, but at a minimum it should be addressed in the discussion.

We agree that this issue is an important one. However, as explained above to reply to a similar point made by referee 1, we believe that the brood cells of the honeycomb are more similar, under the chemical point of view, to gelatin cells than artificial wax cells. In fact, a notable cocoon layer can be found inside any brood cell that has been used at least once by a growing larva (to test this one can melt an old honey comb: what is left is hundreds of little cells made of silk).

As a matter of fact, gelatine is made of collagen proteins and the cocoon is made of silk: another protein of similar polarity.

In any case a comment on that is certainly needed and was added in the new version of the manuscript (please, see lines 80-83 in Main text_NEW_CLEAN file).

At several points through the manuscript the term "sprinkled" is used to describe application of propolis to the capsules (e.g. line 247). This term comes across as overly casual. Please be more descriptive about the application method at line 247 and replace "sprinkled" with "applied" or another verb in the legend for Figure 1.

We now explain how the propolis extract was applied to the capsules used for the experiments (please, see lines 247-251 in Main text_NEW_CLEAN file).

We now use applied throughout the text.

*The paragraph starting on line 166 should be revised to improve readability and to avoid implying that *A. mellifera* has evolved over the last 100+ years to use propolis against varroa mites. The sentence at lines 170-172, is especially problematic in this regard.*

The problematic paragraph has been removed.

In the methods on line 266 it states that "crippled wings" were observed at emergence. How was the "crippled" status of emerging bees determined. Was this purely visual? Provide more details on how this was determined by observers.

We added more details on this in the methods section (please, see lines 270-273 in Main text_NEW_CLEAN file).

Minor points:

Line 44: Remove word "dramatically"

Done.

Lines 73-74: Use consistent units -- either g/kg or mg/g for both

Done.

Line 75: "to reinforce the structure and disinfect the cells" should probably be removed here. Or references should be provided to support this statement.

We removed that part of the sentence.

Line 100: "narcoleptic" is probably not the right word in this context.

We removed "narcoleptic".

Line 108: change "had _not_ effect" to "had _no_ effect"

Done.

Line 120: Change "indicates" to "is consistent with"

Done.

Figure 3: Remove "(adult)" from Fig. 3A

Done.

Line 159: "indicates" is too strong a term as more research is needed. Change to "suggests"
Done

Line 261: Provide the composition of the sugar candy used
Done.

Line 288: Check spelling of spectrophotometer
We corrected the word where the spelling was wrong.

Line 205: Add "the" to "observation in _the_ laboratory"
Done.

Appendix B

Comments to the Author(s).

In general, the authors have greatly improved the presentation of the data in this manuscript. The results are exciting, but I think it is important to further temper the wording and conclusions even more.

The abstract is fine, but for example, line 175 of the conclusion states, "... a significant reduction of the survival and reproduction of .. the mites in brood cells..." While it is true there was a statistically significant reduction in survival and reproduction of the mites, in actuality there was about an 18% reduction in mite survival, and of the mites that survived, 44% were infertile, or less than 8% overall. Relative to mite treatments, or resistance behaviors, these reductions are relatively small (but still exciting!). My suggestions for tempering the wording are as follows:

We sincerely thank the Reviewer for the help in improving our manuscript. Below we show how we dealt with her/his precious suggestions.

As for the need to temper our conclusions, we changed a little bit the wording in the abstract as follows.

- a) We use "potential" instead of "clear" where we speak about the effect on varroa population (this is because the impact on Varroa mortality and fertility is clear and significant but the resulting impact on Varroa population dynamics was not tested here).
- b) We write: "We conclude that propolis can be regarded as a natural pesticide used by the honey bee to limit a dangerous parasite" instead of: "We conclude that propolis is a potent natural pesticide used by the honey bee to limit a dangerous parasite".

1. Line 175: We found an 18% (or whatever actual mean was) reduction in the survival of the mites, and 8% of the surviving mites were infertile, indicating that some reduction in survival and reproduction of mites in brood cells is achieved, with benefits for the bees developing in those cells."

We do not want to claim anything more than what we found and agree with the Reviewer about the need to temper our conclusions.

As from Dataset S1 (which is part of the submission), our results are the following:

	V+P-E+	V+P+E+
Mortality	6.1	18.6
Fertility	46.4	26.2

Therefore, the percentage reduction of survival in propolis treated cells is 13%, which results from the following calculation:

$$(\text{survival in propolis treated cells} - \text{survival in control cells}) / (\text{survival in control cells}) = ((100 - 18.6) - (100 - 6.1)) / (100 - 6.1) = (81.4 - 93.9) / 93.9 = -13\%$$

This figure looks rather comforting because it indicates a rather mild effect on mite mortality so that beekeepers will not be encouraged to think that they may fight the mite with propolis.

Instead, the percentage reduction in fertility in propolis treated cells is 44% which results from the following calculation:

$$(\text{fertility in propolis treated cells} - \text{fertility in control cells}) / (\text{fertility in control cells}) = (26.2 - 46.4) / 46.4 = -44\%$$

This latter figure is actually quite impressive and probably not in line with the suggestion of the referee. Of course, we could say something like "the fertility of surviving mites was reduced TO 26% (instead of reduced BY 44%) but this information would be misleading since we did not

evaluate absolute fertility (this should be done in natural brood cells) but rather the relative effect caused by propolis and the only way to estimate the relative effect is to divide by the fertility observed in the control.

In conclusion, a sentence like the following:

“We found that propolis reduced by 13% the survival of mites, whereas the fertility of surviving mites was reduced to 26%, indicating that some reduction in survival and reproduction of mites in brood cells is achieved, with benefits for the bees developing in those cells.” would sound even more triumphalistic than before.

Therefore, we propose to stick to the original version removing the word “significant” since this word refers both to the statistical quality of results (and the differences we got are statistically significant!) and, in common language, to the relevance of data (and the relevance of our results can be debated, if one expects the extermination of parasites or their fully sterilization). Please, see line 175.

2. The reduction in DWV signs when bees developed in gelatin capsules (Fig 2A), relative to DWV in adult bees fed sugar solution with propolis (Fig S3) is interesting, and hints at the possible mode of action of propolis on virus, which is evidently by volatiles or contact and not by oral inoculation. I think the authors could draw out this comparison in the Discussion, particularly in lines 127-133 and line 158 (which without more explanation to line 158, may appear to contradict lines 127-133 to a casual reader).

Actually, we do not think that propolis has got a direct effect on DWV, and this is why we did not comment about the possible mode of action of propolis on the virus. In other words, we think that propolis on its own cannot do much to the virus and the reduction in the percentage of symptomatic bees among those maintained in propolis treated cells is the result of an indirect effect of propolis: a by-product of the clear negative effect of this substance on the infesting mites, which, instead, are certainly responsible for an increased viral replication.

We believe that there is no contradiction between our statements in lines 127-133 and line 158, because in the first place we exclude a direct effect and rather suggest an indirect effect (i.e. mite mediated) and, in the second place, we report a further evidence of the lack of a direct effect.

On the other hand, a possible contradiction would arise if we accepted the change proposed below, because the sentence suggested by the reviewer would then read: “This suggests that, by impacting the survival and reproduction of mites, propolis may reduce the indirect effects of mite infestation on bees and, in particular, may reduce the activation of viral replication, a common outcome of mite parasitism, and the most important pathogenic effect”. As such the sentence may indicate a possible direct effect of propolis on the virus which we do not support.

Possibly, the misunderstanding depends on the fact that the meaning of “direct” and “indirect effect” is not that clear. Therefore, we propose to change the original sentence in lines 127-133 with:

“The phenotype of mite-infested bees emerging from cells treated with propolis is consistent with a lower DWV load in comparison with parasitized bees from untreated cells, similar to previous observations¹⁶. This suggests that, by impacting the survival and reproduction of mites, propolis may indirectly reduce the mite induced viral replication²².”

3. Line 108: “... propolis does not seem to be able to counteract the negative, direct effects of mite infestation on developing bees..” relative to ?? need to complete sentence.

Done (please, see lines 106-108).

4. Explanation of the survival curves (Figs 2B and 3B): Lines 115 (“Propolis increases survival of adult bees” and 146: (“... but propolis significantly decreased the mortality of mite infested bees... ”), please consider adding, “but not to the level of uninfested bees.”

We changed the first half of the sentence (please, see line 117) but not the second since the suggested change can be misleading as we showed above.

5. line 127: “propolis may reduce the indirect effects... and in particular, may reduce the activation of viral replication.”

Done (please, see lines 127-129).

6. line 31: *Ascospaera apis* is the causative agent of chalkbrood, not stonebrood.

Done (please, see line 31)

some minor English grammar suggestions

line 82: *different from the wax that comprises the brood cells, but does not differ from the silk cocoon layer...*

Done (please, see line 82)